# Towards a better understanding of Vector Quantized Autoencoders

## Abstract

Deep neural networks with discrete latent variables offer the promise of better symbolic reasoning, and learning abstractions that are more useful to new tasks. There has been a surge in interest in discrete latent variable models, however, despite several recent improvements, the training of discrete latent variable models has remained challenging and their performance has mostly failed to match their continuous counterparts. Recent work on vector quantized autoencoders (VQ-VAE) has made substantial progress in this direction, with its perplexity almost matching that of a VAE on datasets such as CIFAR-10. In this work, we investigate an alternate training technique for VQ-VAE, inspired by its connection to the Expectation Maximization (EM) algorithm. Training the discrete autoencoder with EM and combining it with sequence level knowledge distillation alows us to develop a non-autoregressive machine translation model whose accuracy almost matches a strong greedy autoregressive baseline Transformer, while being 3.3 times faster at inference.

## 1 Introduction

Unsupervised learning of meaningful representations is a fundamental problem in machine learning since obtaining labeled data can often be very expensive. Continuous representations have largely been the workhorse of unsupervised deep learning models of images (Goodfellow et al., 2014; van den Oord et al., 2016; Kingma et al., 2016; Salimans et al., 2017; Parmar et al., 2018), audio (Van Den Oord et al., 2016; Reed et al., 2017), and video (Kalchbrenner et al., 2016). However, it is often the case that datasets are more naturally modeled as a sequence of discrete symbols rather than continuous ones. For example, language and speech are inherently discrete in nature and images are often concisely described by language, see e.g., Vinyals et al. (2015). Improved discrete latent variable models could also prove useful for learning novel data compression algorithms (Theis et al., 2017), while having far more interpretable representations of the data.

We build on Vector Quantized Variational Autoencoder (VQ-VAE) (van den Oord et al., 2017), a recently proposed training technique for learning discrete latent variables. The method uses a learned code-book combined with nearest neighbor search to train the discrete latent variable model. The nearest neighbor search is performed between the encoder output and the embedding of the latent code using the $\ell_2$ distance metric. VQ-VAE adopts the standard latent variable model generative process, first sampling latent codes from a prior, $P(z)$, which are then consumed by the decoder to generate data from $P(x \mid z)$. In van den Oord et al. (2017), the authors use both uniform and autoregressive priors for $P(z)$. The resulting discrete autoencoder obtains impressive results on unconditional image, speech, and video generation. In particular, on image generation, VQ-VAE was shown to perform almost on par with continuous VAEs on datasets such as CIFAR-10 (van den Oord et al., 2017). An extension of this method to conditional supervised generation, out-performs continuous autoencoders on WMT English-German translation task (Kaiser et al., 2018).

The work of Kaiser et al. (2018) introduced the Latent Transformer, which set a new state-of-the-art in non-autoregressive Neural Machine Translation. However, additional training heuristics, namely, exponential moving averages (EMA) of cluster assignment counts, and

product quantization (Norouzi & Fleet, 2013) were essential to achieve competitive results with VQ-VAE. In this work, we show that tuning for the code-book size can significantly outperform the results presented in Kaiser et al. (2018). We also exploit VQ-VAE's connection with the expectation maximization (EM) algorithm (Dempster et al., 1977), yielding additional improvements. With both improvements, we achieve a BLEU score of 22.4 on English to German translation, outperforming Kaiser et al. (2018) by 2.6 BLEU. Knowledge distillation (Hinton et al., 2015; Kim & Rush, 2016) provides significant gains with our best models and EM, achieving 26.7 BLEU, which almost matches the autoregressive transformer model with no beam search at 27.0 BLEU, while being 3.3× faster.

Our contributions can be summarized as follows:

1. We show that VQ-VAE from van den Oord et al. (2017) can outperform previous state-of-the-art without product quantization.
2. Inspired by the EM algorithm, we introduce a new training algorithm for training discrete variational autoencoders, that outperforms the previous best result with discrete latent autoencoders for neural machine translation.
3. Using EM training, and combining it sequence level knowledge distillation (Hinton et al., 2015; Kim & Rush, 2016), allows us to develop a non-autoregressive machine translation model whose accuracy almost matches a strong greedy autoregressive baseline Transformer, while being 3.3 times faster at inference.
4. On the larger English-French dataset, we show that *denoising discrete autoencoders* gives us a significant improvement (1.0 BLEU) on top of our non-autoregressive baseline (see Section D).

## 2 VQ-VAE AND THE HARD EM ALGORITHM

The connection between $K$-means, and hard EM, or the Viterbi EM algorithm is well known (Bottou & Bengio, 1995), where the former can be seen a special case of hard-EM style algorithm with a mixture-of-Gaussians model with identity covariance and uniform prior over cluster probabilities. In the following sections we briefly explain the VQ-VAE discrete autoencoder for completeness and it's connection to classical EM.

### 2.1 VQ-VAE DISCRETIZATION ALGORITHM

VQ-VAE models the joint distribution $P_\Theta(x, z)$ where $\Theta$ are the model parameters, $x$ is the data point and $z$ is the sequence of discrete latent variables or codes. Each position in the encoded sequence has its own set of latent codes. Given a data point, the discrete latent code in each position is selected independently using the encoder output. For simplicity, we describe the procedure for selecting the discrete latent code $(z_i)$ in one position given the data point $(x_i)$. The encoder output $z_e(x_i) \in R^D$ is passed through a discretization bottleneck using a nearest-neighbor lookup on embedding vectors $e \in R^{K \times D}$. Here $K$ is the number of latent codes (in a particular position of the discrete latent sequence) in the model. More specifically, the discrete latent variable assignment is given by,

$$z_i = \arg \min_{j \in [K]} \|z_e(x_i) - e_j\|_2 \tag{1}$$

The selected latent variable's embedding is passed as input to the decoder,

$$z_q(x_i) = e_{z_i}$$

The model is trained to minimize:

$$L = l_r + \beta \|z_e(x_i) - \text{sg}(z_q(x_i))\|_2, \tag{2}$$

where $l_r$ is the reconstruction loss of the decoder given $z_q(x)$ (e.g., the cross entropy loss), and, sg (.) is the stop gradient operator defined as follows:

$$\text{sg}(x) = \begin{cases} x & \text{forward pass} \\ 0 & \text{backward pass} \end{cases}$$

To train the embedding vectors $e \in R^{K \times D}$, van den Oord et al. (2017) proposed using a gradient based loss function

$$\|\text{sg}\left(z_e(x_i)\right) - z_q(x_i)\|_2, \tag{3}$$

and also suggested an alternate technique of training the embeddings: by maintaining an exponential moving average (EMA) of all the encoder hidden states that get assigned to it. It was observed in Kaiser et al. (2018) that the EMA update for training the code-book embedding, results in more stable training than using gradient-based methods. We analyze this in more detail in Section 5.1.1.

Specifically, an exponential moving average is maintained over the following two quantities: 1) the embeddings $e_j$ for every $j \in [1, \ldots, K]$ and, 2) the count $c_j$ measuring the number of encoder hidden states that have $e_j$ as it's nearest neighbor. The counts are updated in a mini-batch of targets as:

$$c_j \leftarrow \lambda c_j + (1 - \lambda) \sum_i \mathbb{1}\left[z_q(x_i) = e_j\right], \tag{4}$$

with the embedding $e_j$ being subsequently updated as:

$$e_j \leftarrow \lambda e_j + (1 - \lambda) \sum_i \frac{\mathbb{1}\left[z_q(x_i) = e_j\right] z_e(x_i)}{c_j}, \tag{5}$$

where $\mathbb{1}[.]$ is the indicator function and $\lambda$ is a decay parameter which we set to 0.999 in our experiments. This amounts to doing stochastic gradient in the space of both code-book embeddings and cluster assignments. These techniques have also been successfully used in minibatch $K$-means (Sculley, 2010) and online EM (Liang & Klein, 2009; Sato & Ishii, 2000).

The generative process for our latent variable NMT model, $P(y, z \mid x)$, begins by *autoregressively* sampling a sequence of discrete latent codes from a model conditioned on the input $x$,

$$P(z \mid x) = \prod_{i=1}^{|z|} P\left(z_i \mid z_{1,\ldots,(i-1)}, x\right), \tag{6}$$

which we refer to as the Latent Predictor model (Kaiser et al., 2018). The decoder then consumes this sequence of discrete latent variables to generate the target $y$ all at once, where

$$P(y \mid z, x) = \prod_{j=1}^{|y|} P(y_j \mid z, x). \tag{7}$$

The autoregressive learned prior prior is fit on the discrete latent variables produced by the autoencoder. Our goal is to learn a sequence of latents, that is *much shorter* than the targets, $|z| \ll |y|$, thereby speeding up decoding significantly with no loss in accuracy. The architecture of the encoder, the decoder, and the latent predictor model are described in further detail in Section 5.

## 2.2 Hard EM and the $K$-means algorithm

In this section we briefly recall the hard Expectation maximization (EM) algorithm (Dempster et al., 1977). Given a set of data points $(x_1, \ldots, x_N)$, the hard EM algorithm approximately solves the following optimization problem:

$$\Theta^* = \arg\max_{\Theta} \max_{z_1, \ldots, z_N} P_{\Theta}(x_1, \ldots, x_N, z_1, \ldots, z_N), \tag{8}$$

Hard EM performs coordinate descent over the following two coordinates: the model parameters $\Theta$, and the hidden variables $z_1, \ldots, z_N$. In other words, hard EM consists of repeating the following two steps until convergence:

1. **E step:** $(z_1, \ldots, z_N) \leftarrow \arg\max_{z_1, \ldots, z_N} P_{\Theta}(x_1, \ldots, x_N, z_1, \ldots, z_N)$,

2. **M step:** $\Theta \leftarrow \arg\max_\Theta P_\Theta(x_1, \ldots, x_N, z_1, \ldots, z_N)$

A special case of the hard EM algorithm is $K$-means clustering (MacQueen et al., 1967; Bottou & Bengio, 1995) where the likelihood is modelled by a Gaussian with identity covariance matrix. Here, the means of the $K$ Gaussians are the parameters to be estimated,

$$\Theta = \langle \mu^1, \ldots, \mu^K \rangle, \quad \mu^k \in R^D.$$

With a uniform prior over the hidden variables ($P_\Theta(z_i) = \frac{1}{K}$), the marginal is given by $P_\Theta(x_i \mid z_i) = \mathcal{N}(\mu^{z_i}, I)(x_i)$. In this case, equation (8) is equivalent to:

$$\left(\mu^1, \ldots, \mu^K\right)^* = \arg\max_{\mu^1, \ldots, \mu^K} \min_{z_1, \ldots, z_N} \sum_{i=1}^N \|\mu^{z_i} - x_i\|_2^2 \tag{9}$$

Note that optimizing equation (9) is NP-hard, however one can find a local optima by applying coordinate descent until convergence:

1. **E step:** Cluster assignment is given by,

$$z_i \leftarrow \arg\min_{j \in [K]} \left\|\mu^j - x_i\right\|_2^2, \tag{10}$$

2. **M step:** The means of the clusters are updated as,

$$c_j \leftarrow \sum_{i=1}^N \mathbb{1}[z_i = j]; \quad \mu^j \leftarrow \frac{1}{c_j} \sum_{i=1}^N \mathbb{1}[z_i = j] x_i. \tag{11}$$

We can now easily see the connections between the training updates of VQ-VAE and $K$-means clustering. The encoder output $z_e(x) \in R^D$ corresponds to the data point while the discrete latent variables corresponds to clusters. Given this, Equation 1 is equivalent to the E-step (Equation 10) and the EMA updates in Equation 4 and Equation 5 converge to the M-step (Equation 11) in the limit. The M-step in $K$-means overwrites the old values while the EMA updates interpolate between the old values and the M step update.

## 3 VQ-VAE TRAINING WITH EM

In this section, we investigate a new training strategy for VQ-VAE using the EM algorithm.

### 3.1 EXPECTATION MAXIMIZATION

First, we briefly describe the EM algorithm. While the hard EM procedure selects one cluster or latent variable assignment for a data point, here the data point is assigned to a mixture of clusters. Now, the optimization objective is given by,

$$\Theta^* = \arg\max_\Theta P_\Theta(x_1, \ldots, x_N)$$
$$= \arg\max_\Theta \sum_{z_1, \ldots, z_N} P_\Theta(x_1, \ldots, x_N, z_1, \ldots, z_N)$$

Coordinate descent algorithm is again used to approximately solve the above optimization algorithm. The E and M step are given by:

1. **E step:**

$$\rho(z_i) \leftarrow P_\Theta(z_i \mid x_i), \tag{12}$$

2. **M step:**

$$\Theta \leftarrow \arg\max_\Theta \mathbb{E}_{z_i \sim \rho}[\log P_\Theta(x_i, z_i)] \tag{13}$$

## 3.2 Vector Quantized Autoencoders trained with EM

Now, we describe vector quantized autoencoders training using the EM algorithm. As discussed in the previous section, the encoder output $z_e(x) \in R^D$ corresponds to the data point while the discrete latent variables corresponds to clusters. The E step instead of hard assignment now produces a probability distribution over the set of discrete latent variables (Equation 12). Following VQ-VAE, we continue to assume a uniform prior over clusters, since we observe that training the cluster priors seemed to cause the cluster assignments to collapse to only a few clusters. The probability distribution is modeled as a Gaussian with identity covariance matrix,

$$P_\Theta(z_i \mid z_e(x_i)) \propto e^{-\|e_{z_i} - z_e(x_i)\|_2^2}$$

As an alternative to computing the full expectation term in the M step (Equation 13) we perform Monte-Carlo Expectation Maximization (Wei & Tanner, 1990) by drawing $m$ samples $z_i^1, \cdots, z_i^m \sim \text{Multinomial}\left(-\|e_1 - z_e(x_i)\|_2^2, \ldots, -\|e_K - z_e(x_i)\|_2^2\right)$, where $\text{Multinomial}(l_1, \ldots, l_K)$ refers to the $K$-way multinomial distribution with logits $l_1, \ldots, l_K$. This results in a less diffuse target for the autoregressive prior. Thus, the E step can be finally written as:

$$\textbf{E step:} \qquad z_i^1, \ldots, z_i^m \leftarrow \text{Multinomial}\left(-\|e_1 - z_e(x_i)\|_2^2, \ldots, -\|e_K - z_e(x_i)\|_2^2\right)$$

The model parameters $\Theta$ are then updated to maximize this Monte-Carlo estimate in the M step given by

$$\textbf{M step:} \qquad c_j \leftarrow \frac{1}{m} \sum_{i=1}^N \sum_{l=1}^m \mathbb{1}\left[z_i^l = j\right]; \qquad e_j \leftarrow \frac{1}{mc_j} \sum_{i=1}^N \sum_{l=1}^m \mathbb{1}\left[z_i^l = j\right] z_e(x_i).$$

Instead of exactly following the above M step update, we use the EMA version of this update similar to the one described in Section 2.1.

When sending the embedding of the discrete latent to the decoder, instead of sending the posterior mode, $\text{argmax}_z P(z \mid x)$, similar to hard EM and $K$-means, we send the average of the embeddings of the sampled latents:

$$z_q(x_i) = \frac{1}{m} \sum_{l=1}^m e_{z_i^l}. \tag{14}$$

Since $m$ latent code embeddings are sent to the decoder in the forward pass, all of them are updated in the backward pass for a single training example. In hard EM training, only one of them is updated during training. Sending averaged embeddings also results in more stable training using the EM algorithm compared to VQ-VAE as shown in Section 5.

To train the latent predictor model (Section 2.1) in this case, we use an approach similar to *label smoothing* (Pereyra et al., 2017): the latent predictor model is trained to minimize the cross entropy loss with the labels being the average of the one-hot labels of $z_i^1, \ldots, z_i^m$.

## 4 Other Related Work

Variational autoencoders were first introduced by Kingma & Welling (2014) for training continuous representations; unfortunately, training them for discrete latent variable models has proved challenging. One promising approach has been to use various gradient estimators for discrete latent variable models, starting with the REINFORCE estimator of Williams (1992), an unbiased, high-variance gradient estimator. Subsequent work on improving the variance of the REINFORCE estimator are REBAR (Tucker et al., 2017) and RELAX (Grathwohl et al., 2017). An alternate approach towards gradient estimators is to use continuous relaxations of categorical distributions, for e.g., the Gumbel-Softmax reparametrization trick (Jang et al.,

2016; Maddison et al., 2016). These methods provide biased but low variance gradients for training.

Machine translation using deep neural networks have been shown to achieve impressive results (Sutskever et al., 2014; Bahdanau et al., 2014; Cho et al., 2014; Vaswani et al., 2017). The state-of-the-art models in Neural Machine Translation are all auto-regressive, which means that during decoding, the model consumes all previously generated tokens to predict the next one. Recently, there have been multiple efforts to speed-up machine translation decoding. Gu et al. (2017) attempts to address this issue by using the Transformer model (Vaswani et al., 2017) together with the REINFORCE algorithm (Williams, 1992), to model the *fertilities* of words. The main drawback of the approach of Gu et al. (2017) is the need for extensive fine-tuning to make policy gradients work, as well as the non-generic nature of the solution. Lee et al. (2018) propose a non-autoregressive model using iterative refinement. Here, instead of decoding the target sentence in one-shot, the output is successively refined to produce the final output. While the output is produced in parallel at each step, the refinement steps happen sequentially.

## 5 EXPERIMENTS

In this section we report our experiments with VQ-VAE and EM on the English-German translation task, with the aim of improving the decoding speed of autoregressive translation models. Our model and generative process follows the architecture proposed in Kaiser et al. (2018) and is depicted in Figure 1. For all our experiments, we use the Adam (Kingma & Ba, 2014) optimizer and decay the learning rate exponentially after initial warm-up steps. Unless otherwise stated, the dimension of the hidden states of the encoder and the decoder is 512, see Table 4 for a comparison of models with lower dimension. For all configurations we select the optimal hyperparameters by using WMT'13 English-German as the validation set and reporting the BLEU score on the WMT'14 English-German test set.

### 5.1 MACHINE TRANSLATION

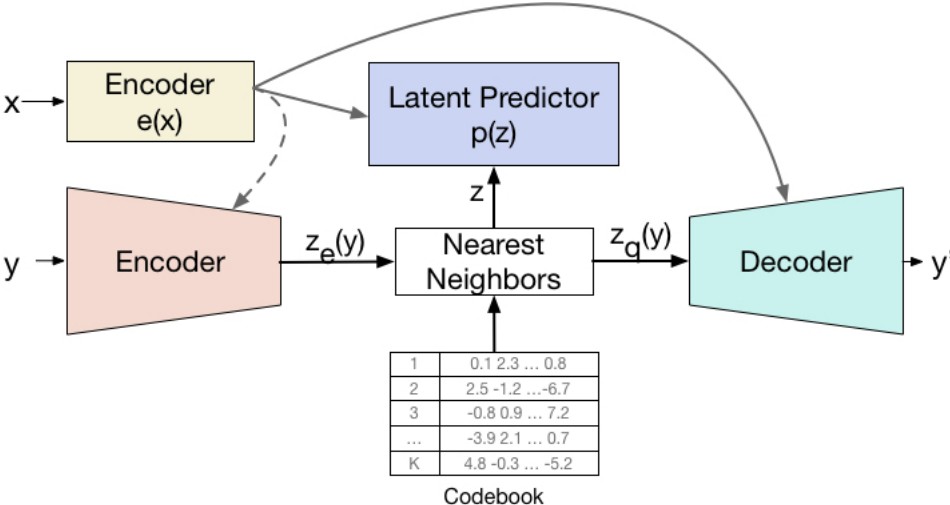

Figure 1: VQ-VAE model adapted to conditional supervised translation as described in Kaiser et al. (2018). We use $x$ and $y$ to denote the source and target sentence respectively. The encoder, the decoder and the latent predictor now additionally condition on the source sentence $x$.

In Neural Machine Translation with latent variables, we model $P(y, z \mid x)$, where $y$ and $x$ are the target and source sentence respectively. Our model architecture, depicted in Figure 1, is similar to the one in Kaiser et al. (2018). The encoder function is a series of strided convolutional layers with residual convolutional layers in between and takes target sentence $y$

as input. The source sentence $x$ is converted to a sequence of hidden states through multiple causal self-attention layers. In Kaiser et al. (2018), the encoder of the autoencoder attends additionally to this sequence of continuous representation of the source sentence. We use VQ-VAE as the discretization algorithm. The decoders, applied after the bottleneck layer uses transposed convolution layers whose continuous output is fed to a transformer decoder with causal attention, which generates the output.

The results are summarized in Table 1. Our implementation of VQ-VAE achieves a significantly better BLEU score and faster decoding speed compared to Kaiser et al. (2018). We found that tuning the code-book size (number of clusters) for using $2^{12}$ discrete latents achieves the best accuracy which is 16 times smaller as compared to the code-book size in Kaiser et al. (2018). Additionally, we see a large improvement in the performance of the model by using sequence-level distillation (Hinton et al., 2015; Kim & Rush, 2016), as has been observed previously in non-autoregressive models (Gu et al., 2017; Lee et al., 2018). Our teacher model is a base Transformer (Vaswani et al., 2017) that achieves a BLEU score of 28.1 and 27.0 on the WMT'14 test set using beam search decoding and greedy decoding respectively. The distilled data is decoded from the base Transformer using a beam size of 4. Our VQ-VAE model trained with soft EM and distillation, achieves a BLEU score of 26.7, without noisy parallel decoding (Gu et al., 2017). This perforamce is 1.4 bleu points lower than an autoregressive model decoded with a beam size of 4, while being $4.1\times$ faster. Importantly, we nearly match the same autoregressive model with beam size 1 (greedy decoding), with a $3.3\times$ speedup.

The length of the sequence of discrete latent variables is shorter than that of target sentence $y$. Specifically, at each compression step of the encoder we reduce its length by half. We denote by $n_c$, the compression factor for the latents, i.e. the number of steps for which we do this compression. In almost all our experiments, we use $n_c = 3$ reducing the length by 8. We can decrease the decoding time further by increasing the number of compression steps. As shown in Table 1, by setting $n_c$ to 4, the decoding time drops to 58 milliseconds achieving 25.4 BLEU while a NAT model (Gu et al., 2017) with similar decoding speed achieves only 18.7 BLEU. Note that, all NAT models also train with sequence level knowledge distillation from an autoregressive teacher.

### 5.1.1 ANALYSIS

**Attention to Source Sentence Encoder:** While the encoder of the discrete autoencoder in Kaiser et al. (2018) attends to the output of the encoder of the source sentence, we find that to be unnecessary, with both models achieving the same BLEU score with $2^{12}$ latents. Removing this attention step results in more stable training (see Figure 3) and is the *main reason* why VQ-VAE works in our setup (see Table 1) without the use of Product Quantization (DVQ) (Kaiser et al., 2018). Note that the decoder of the discrete autoencoder in both Kaiser et al. (2018) and our work does not attend to the source sentence.

**Size of Discrete Latent Variable code-book:** Table 2 shows the BLEU score for different code-book sizes for models trained using VQ-VAE without distillation. While Kaiser et al. (2018) use $2^{16}$ as their code-book size, we find that $2^{12}$ gives the best performance.

**Number of samples in Monte-Carlo EM update:** While training with EM, we perform a Monte-Carlo update with a small number of samples (Section 3.2). Table 3 shows the impact of number of samples on the final BLEU score.

**VQ-VAE vs Other Discretization Techniques:** We compare the Gumbel-Softmax of (Jang et al., 2016; Maddison et al., 2016) and the improved semantic hashing discretization technique proposed in Kaiser et al. (2018) to VQ-VAE. When trained with sequence level knowledge distillation, the model using Gumbel-Softmax reached 23.2 BLEU, the model using improved semantic hashing reached 24.1 BLEU, and the model using VQ-VAE reached 26.4 BLEU on WMT'14 English-German.

| Model | $n_c$ | $n_s$ | BLEU | Latency | Speedup |
|---|---|---|---|---|---|
| Autoregressive Model (beam size=4) | - | - | 28.1 | 331 ms | $1\times$ |
| Autoregressive Baseline (no beam-search) | - | - | 27.0 | 265 ms | $1.25\times$ |
| NAT + distillation | - | - | 17.7 | 39 ms | $15.6\times$ [*] |
| NAT + distillation + NPD=10 | - | - | 18.7 | 79 ms | $7.68\times$ [*] |
| NAT + distillation + NPD=100 | - | - | 19.2 | 257 ms | $2.36\times$ [*] |
| LT + Semhash | - | - | 19.8 | 105 ms | $3.15\times$ |
| Our Results | | | | | |
| VQ-VAE | 3 | - | 21.4 | 81 ms | $4.08\times$ |
| VQ-VAE with EM | 3 | 5 | 22.4 | 81 ms | $4.08\times$ |
| VQ-VAE + distillation | 3 | - | 26.4 | 81 ms | $4.08\times$ |
| VQ-VAE with EM + distillation | 3 | 10 | **26.7** | 81 ms | $4.08\times$ |
| VQ-VAE with EM + distillation | 4 | 10 | 25.4 | 58 ms | $5.71\times$ |

Table 1: BLEU score and decoding times for different models on the WMT'14 English-German translation dataset. The baseline is the autoregressive Transformer of Vaswani et al. (2017) with no beam search, NAT denotes the Non-Autoregressive Transformer of Gu et al. (2017), and LT + Semhash denotes the Latent Transformer from van den Oord et al. (2017) using the improved semantic hashing discretization technique of Kaiser & Bengio (2018). NPD refers to *noisy parallel decoding* as described in Gu et al. (2017). We use the notation $n_c$ to denote the compression factor for the latents, and the notation $n_s$ to denote the number of samples used to perform the Monte-Carlo approximation of the EM algorithm. Distillation refers to sequence level knowledge distillation from Hinton et al. (2015); Kim & Rush (2016). We used a code-book of size $2^{12}$ for VQ-VAE (for with and without EM) with a hidden dimension of size 512. Decoding is performed on a single CPU machine with an NVIDIA GeForce GTX 1080 with a batch size of 1

[*] Speedup reported for these items are compared to the decode time of 408 ms for an autoregressive Transformer from Gu et al. (2017).

## 6 Conclusion

We investigate an alternate training technique for VQ-VAE inspired by its connection to the EM algorithm. Training the discrete autoencoder with EM and combining it with sequence level knowledge distillation, allows us to develop a non-autoregressive machine translation model whose accuracy almost matches a greedy autoregressive baseline, while being 3.3 times faster at inference. While sequence distillation is very important for training our best model, we find that the improvements from EM on harder tasks is quite significant. We hope that our results will inspire further research on using vector quantization for fast decoding of autoregressive sequence models.

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

## A   Image Reconstruction

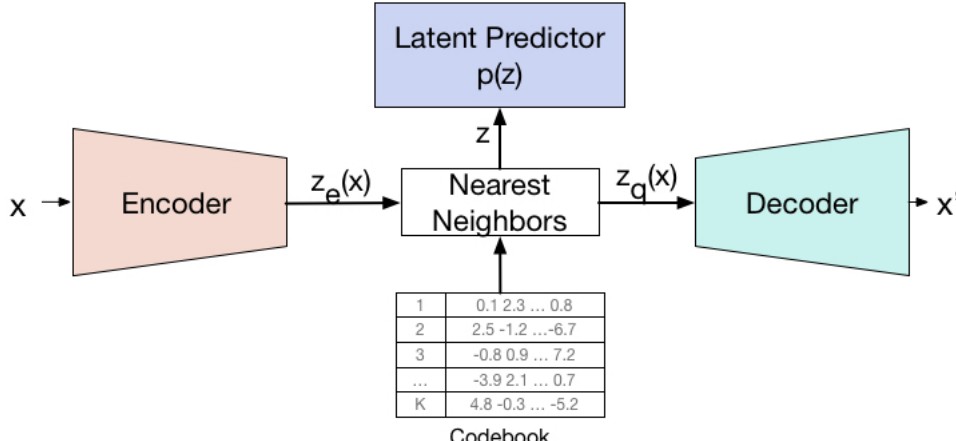

Figure 2: VQ-VAE model as described in van den Oord et al. (2017) for image reconstruction. We use the notation $x$ to denote the input image, with the output of the encoder $z_e(x) \in R^D$ being used to perform nearest neighbor search to select the (sequence of) discrete latent variable. The selected discrete latent is used to train the latent predictor model, while the embedding $z_q(x)$ of the selected discrete latent is passed as input to the decoder.

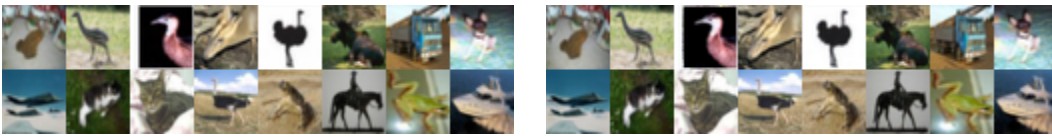

Figure 3: Samples of original and reconstructed images from CIFAR-10 using EM trained with a code-book of size $2^8$.

In this section we report additional experiments we performed using VQ-VAE and EM for the task of image reconstruction. We train a discrete autoencoder with VQ-VAE (van den Oord et al., 2017) and EM on the CIFAR-10 data set, modeling the joint probability $P(x, z)$, where $x$ is the image and $z$ are the discrete latent codes. We use a field of $8 \times 8 \times 10$ latents with a code-book of size $2^8$ each containing 512 dimensions. We maintain the same encoder and decoder as used in Machine Translation. For the encoder, we use 4 convolutional layers, with kernel size $5 \times 5$ and strides $2 \times 2$, followed by 2 residual layers, and a single dense layer. For the decoder, we use a single dense layer, 2 residual layers, and 4 deconvolutional layers. Figure 3 shows that our reconstructions are on par with hard EM training.

We also train discrete autoencoders on the SVHN dataset (Netzer et al., 2011), with both VQ-VAE (van den Oord et al., 2017) and EM. The autoencoder is similar to our CIFAR-10 model, where each $n_x = 32 \times 32 \times 3$ image is encoded into 640 discrete latents from a shared codebook of size 256. By contrasting the reconstructions from several training runs for VQ-VAE (left) and EM (right), we find that training with EM is more reliable and the reconstructions are of high quality (Figure 4)

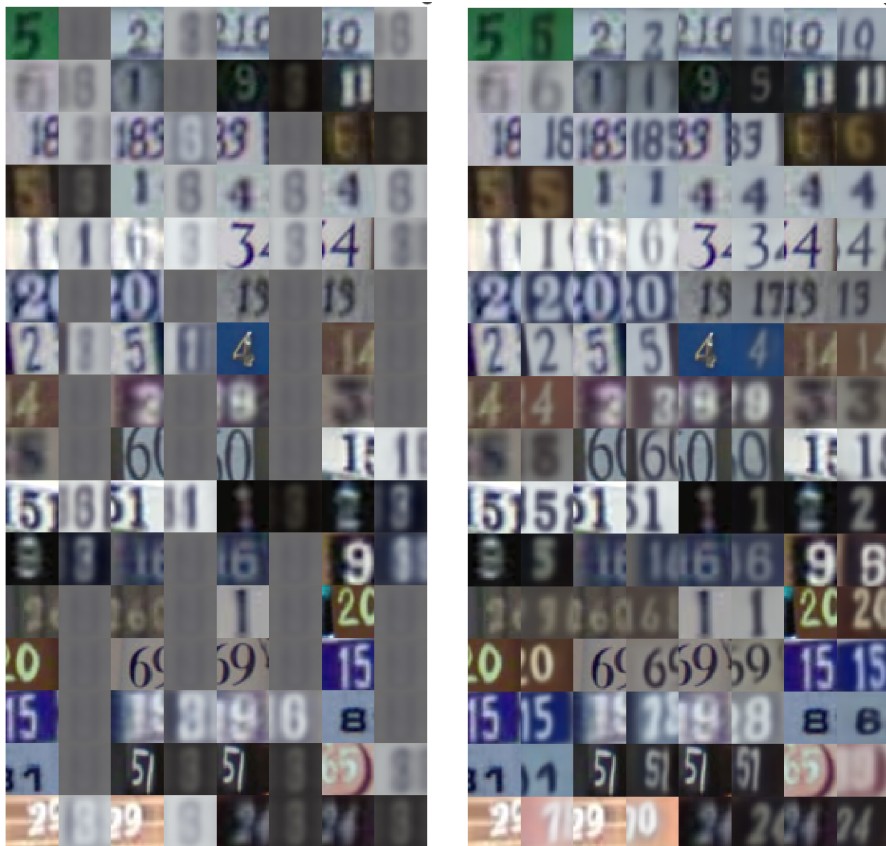

Figure 4: On the left are reconstructions from a model trained with VQ-VAE (van den Oord et al., 2017) and the right figure shows reconstructions from EM training, our approach.

## B    ABLATION TABLES

| Model | Code-book size | BLEU |
|-------|----------------|------|
| VQ-VAE | $2^{10}$ | 20.8 |
| VQ-VAE | $2^{12}$ | 21.6 |
| VQ-VAE | $2^{14}$ | 21.0 |
| VQ-VAE | $2^{16}$ | 21.8 |

Table 2: Results showing the impact of the discrete vocabulary on the BLEU score for the WMT'14 English-German dataset. The hidden dimension is 512 for all runs.

| Model | $n_c$ | $n_s$ | BLEU | Latency | Speedup |
|---|---|---|---|---|---|
| VQ-VAE with EM + distillation | 3 | 1 | 25.8 | 81 ms | 4.08× |
| VQ-VAE with EM + distillation | 3 | 5 | 26.4 | 81 ms | 4.08× |
| VQ-VAE with EM + distillation | 3 | 10 | **26.7** | 81 ms | 4.08× |
| VQ-VAE with EM + distillation | 3 | 25 | 26.6 | 81 ms | 4.08× |
| VQ-VAE with EM + distillation | 3 | 50 | 26.5 | 81 ms | 4.08× |
| VQ-VAE with EM + distillation | 3 | 100 | 25.8 | 81 ms | 4.08× |
| VQ-VAE with EM + distillation | 4 | 1 | 24.1 | 58 ms | 5.71× |
| VQ-VAE with EM + distillation | 4 | 5 | 24.7 | 58 ms | 5.71× |
| VQ-VAE with EM + distillation | 4 | 10 | 25.4 | 58 ms | 5.71× |
| VQ-VAE with EM + distillation | 4 | 25 | 25.1 | 58 ms | 5.71× |
| VQ-VAE with EM + distillation | 4 | 50 | 23.6 | 58 ms | 5.71× |
| VQ-VAE with EM + distillation | 4 | 100 | 24.8 | 58 ms | 5.71× |

Table 3: Results showing the impact of number of samples used to perform the Monte-Carlo EM update on the BLEU score for the WMT'14 English-German dataset. The codebook size for all runs in this table is $2^{12} \times 512$.

| Model | Hidden dimension | $n_s$ | BLEU | Latency | Speedup |
|---|---|---|---|---|---|
| VQ-VAE + distillation | 256 | - | 24.5 | 76 ms | 4.36× |
| VQ-VAE with EM + distillation | 256 | 10 | 21.9 | 76 ms | 4.36× |
| VQ-VAE with EM + distillation | 256 | 25 | 25.8 | 76 ms | 4.36× |
| VQ-VAE + distillation | 384 | - | 25.6 | 80 ms | 4.14× |
| VQ-VAE with EM + distillation | 384 | 10 | 22.2 | 80 ms | 4.14× |
| VQ-VAE with EM + distillation | 384 | 25 | 26.2 | 80 ms | 4.14× |

Table 4: Results showing the impact of the dimension of the word embeddings and the hidden layers of the model on the BLEU score for the WMT'14 English-German dataset with a discrete vocabulary of size $2^{12}$.

## C  ADDITIONAL ANALYSIS

**Gradient based update vs EMA update of code-book:**  The original VQ-VAE paper (van den Oord et al., 2017) proposed a gradient based update rule for learning the code-book where the code-book entries are trained by minimizing $\|\mathrm{sg}\left(z_e(x)\right) - z_q(x)\|_2$. However, it was found in Kaiser et al. (2018) that the EMA update worked better than this gradient based loss. Note that if the gradient based loss was minimized using SGD then the update rule for the embeddings is

$$e_j \leftarrow (1 - \eta)e_j + \eta \left( \frac{\sum_i \mathbb{1}\left[z_q(x_i) = e_j\right] z_e(x_i)}{\sum_i \mathbb{1}\left[z_q(x_i) = e_j\right]} \right), \tag{15}$$

for a learning rate $\eta$. This is quite similar to the EMA update rule of Equation 5, with the only difference being that the latter also maintains an EMA over the counts $c_j$. When using SGD with momentum or Adam, the update rule becomes quite different however, since we now take the moving average of the gradient term itself, before subtracting it from current value of the embedding $e_j$. This is similar to the issue of using weight decay with Adam, where using the $\ell_2$ penalty in the loss function results in worse performance (Loshchilov & Hutter, 2017).

**Model Size:**  The effect of model size on BLEU score for models trained with EM and distillation is shown in Table 4.

**Robustness of EM to Hyperparameters:** While EM training gives a small performance improvement, we find that it also leads to more robust training for machine translation.

Our experiments on image reconstruction on SVHN (Netzer et al., 2011) in section A also highlight the robustness of EM training. The training approach from van den Oord et al. (2017) exhibits high variance on reconstruction quality, while EM is much more stable, resulting in good reconstructions in almost all training runs.

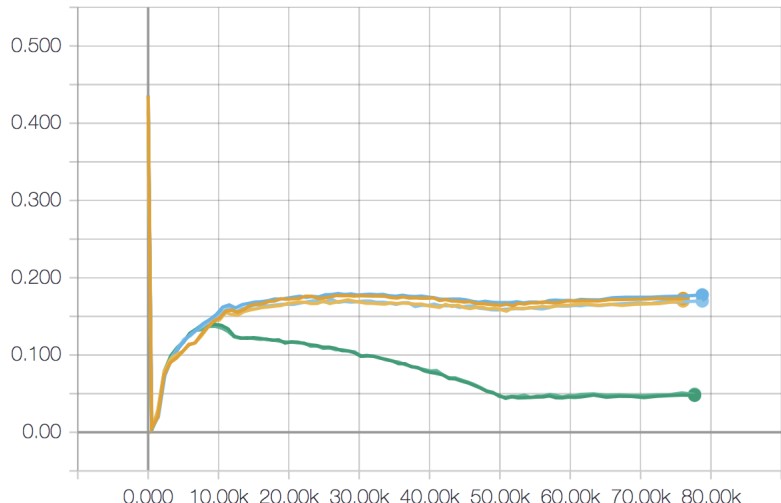

Figure 5: Comparison of VQ-VAE (green curve) vs EM with different number of samples (yellow and blue curves) on the WMT'14 English-German translation dataset with a codebook size of $2^{14}$, with the encoder of the discrete autoencoder attending to the output of the encoder of the source sentence as in Kaiser et al. (2018). The $y$-axis denotes the teacher-forced BLEU score on the test set, which is used only for evaluation while training. Notice that the VQ-VAE run collapsed (green curve), while the EM runs (yellow and blue curves) exhibit more stability.

**Emergence of EOS/PAD latent:** We observe that all the latent sentences for a specific experiment with VQ-VAE or EM end with a fixed latent indicating the end of the sequence. Since we always fix the length of the latent sentence to be $2^{n_c}$ times smaller than the true sentence, the model learns to pad the remainder of the latent sequence with this special code (see Table 5 for examples). Note that one can speed up decoding even further by stopping the Latent Predictor (LP) model as soon as it outputs this special code.

| |
|---|
| 7 89 517 3773 760 760 760 760 |
| 607 1901 1901 3051 760 760 760 760 |
| 2388 15 850 2590 760 760 760 760 |
| 670 127 17 3773 760 760 760 760 |
| 2335 26 129 2986 760 760 760 760 |
| 10 45 1755 766 760 760 760 760 |
| 3773 1082 13 91 760 760 760 760 |
| 1790 38 270 554 760 760 760 760 |
| 2951 2015 91 2418 760 760 760 760 |
| 2951 27 760 760 760 760 760 760 |
| 463 201 3410 3051 760 760 760 760 |

Table 5: Example latent codes for sentences from the WMT'14 English-German dataset highlighting the emergence of the EOS/PAD latent (760 in this case).

**Denoising autoencoder:** We also use *word dropout* with a dropout rate of 0.3 and *word permutation* with a shuffle rate of 0.5 as in Lample et al. (2018). On the WMT English-German we did not notice any improvement from using these regularization techniques, but on the larger WMT English-French dataset, we observe that using a denoising autoencoder significantly improves performance with a gain of 1.0 BLEU on VQ-VAE and 0.9 BLEU over EM (see Table 6).

**Additional analysis on latents:** In order to compute correlations between the discrete latents and $n$-grams in the original text, we computed Point-wise Mutual Information (PMI) and *tf-idf* scores where the latents are treated as documents. However, we were unable to see any semantic patterns that stood out in this analysis.

## D    PRELIMINARY RESULTS ON ENGLISH FRENCH

In this section we report preliminary results on the WMT English-French dataset *without* using knowledge distillation from an autoregressive teacher (Hinton et al., 2015; Kim & Rush, 2016). We use a Transformer base model from Vaswani et al. (2017). Our best non-autoregressive base model trained on non-distilled targets gets 30.0 BLEU compared to the autoregressive base model with the same choice of hyperparameters, which gets 33.3 BLEU (see Table 6). As in the case of English-German, we anticipate that using knowledge distillation Hinton et al. (2015) will likely close this gap.

| Model | $n_c$ | $n_s$ | Code-book size | BLEU | Latency | Speedup |
|---|---|---|---|---|---|---|
| Autoregressive Baseline | - | - | - | 33.3 | 771 ms | 1× |
| Our Results | | | | | | |
| VQ-VAE | 3 | - | 12 | 29.0 | 215 ms | 3.58× |
| VQ-VAE with EM | 3 | 10 | 12 | 29.2 | 215 ms | 3.58× |
| VQ-VAE with reg. | 3 | - | 12 | **30.0** | 215 ms | 3.58× |
| VQ-VAE with EM, reg. | 3 | 10 | 12 | 29.9 | 215 ms | 3.58× |
| VQ-VAE with reg. | 3 | - | 14 | 29.0 | 228 ms | 3.38× |
| VQ-VAE with EM, reg. | 3 | 10 | 14 | 29.5 | 228 ms | 3.38× |

Table 6: BLEU score and decoding times for different models on the WMT'13 English-French translation dataset. The baseline is the autoregressive Transformer of Vaswani et al. (2017) with no beam search, We use the notation $n_c$ to denote the compression factor for the latents, and the notation $n_s$ to denote the number of samples used to perform the Monte-Carlo approximation of the EM algorithm. Reg. refers to *word dropout* with rate 0.3 and *word permutation* with shuffle rate 0.5 as described in Section C. The hidden dimension of the codebook is 512. Decoding is performed on a single CPU machine with an NVIDIA GeForce GTX 1080 with a batch size of 1.

