# OpenReview forum: "Towards a better understanding of Vector Quantized Autoencoders"
_ICLR.cc/2019/Conference_

### Official Review · AnonReviewer3 · 2018-10-26
**Experimental section**

**Rating:** 6
**Confidence:** 3

**Review:**

This paper introduces a new way of interpreting the VQ-VAE,
and proposes a new training algorithm based on the soft EM clustering.

I think the technical aspect of this paper is written concisely.
Introducing the interpretation as hard EM seems natural for me, and the extension
to the soft EM training is sound reasonable.
Mathematical complication is limited, this is also a plus for many non-expert readers.

I'm feeling difficulties in understanding the experimental part.
To be honest, I think the experimental section is highly unorganized, not a quality for ICLR submission.
I'm just wondering why this happens, given clean and organized technical sections...

First, I'm confusing what is the main competent in the Table 1.
In the last paragraph of the page 6, it reads;
"Our implementation of VQ-VAE achieves a significantly better BLEU score and faster decoding speed compared to (10)."
However, Ref. (10) is not mentioned in the Table 1. Which BLEU is the score of Ref. (10)?

Second, terms "VQ-VAE", (soft?)"EM" and "our {model, approach}" are used in a confusing manner.
For example, in Table 1, below the row "Our Results", there are:
- VQ-VAE
- VQ-VAE with EM
- VQ-VAE + distillation
- VQ-VAE with EM + distillation

The "VQ-VAE" is not the proposed model, correct?
My understanding is that the proposal is a VQ-VAE solved via soft EM, which corresponds to "VQ-VAE with EM".

Third, a paragraph "Robustness of EM to Hyperparameters" is mis-leading.
The figure 3 does not show the robustness against a hyperparameter.
It shows the BLEU against the number of "samples" (in fact, there is no explanation about what the "samples" means).
I think hyperparameters are model constants such as the learning rate of the SGD, alpha-beta params for Adam, dimension of hidden units, number of layers, etc. The number of samples are not considered as a model hyperparameter; it's a dataset property.
The figure 5 shows the reconstructed images of the original VQ-VAE and the proposed VQ-VAE with EM.
However, there is no explanation which hyperparameter is tested to assess "the robustness to hyperparameters".

Fourth, there is no experimental report on the image reconstructions (with CIFAR and SVHN) in the main manuscript.
In fact, there is a short paragraph that mentions about the SVHN results,
but it only refers to the appendix.
I think appendix is basically used for additional results or proofs, that are not essential for the main message of the paper.
However, performance in the image reconstruction is one of the main claims written in the abstract, the intro, etc.
So, the authors should include the image reconstruction results in the main body of the paper.
Otherwise, claims about the image reconstructions should be removed from the abstract, etc.


+ Insightful understanding of the VQ-VAE as hard EM clustering
+ Natural and reasonable extension to soft-EM based training of the VQ-VAE
-- Unorganized experiment section. This simply ruins the quality of the technical part.


## after feedback

Some of my concerns are addressed the feedback.
Considering the interesting technical parts, I raise the score upward, to the positive side.

---

> ### Author Response · Authors · 2018-11-09
> **Reply to Reviewer 3**
>
> We thank the reviewer for reading our paper. Below we address specific points raised by the reviewer:
>
> >>>
> I'm feeling difficulties in understanding the experimental part.
> To be honest, I think the experimental section is highly unorganized, not a quality for ICLR submission.
> I'm just wondering why this happens, given clean and organized technical sections...
> >>>
>
> We have made an effort to clean up the experimental section part in the updated draft. We would appreciate specific comments to help us make the experimental section more readable and organized.
>
> >>>
> First, I'm confusing what is the main competent in the Table 1.
> In the last paragraph of the page 6, it reads;
> "Our implementation of VQ-VAE achieves a significantly better BLEU score and faster decoding speed compared to (10)."
> However, Ref. (10) is not mentioned in the Table 1. Which BLEU is the score of Ref. (10)?
> >>>
>
> This should be fixed in the updated version.
>
> >>>
> Second, terms "VQ-VAE", (soft?)"EM" and "our {model, approach}" are used in a confusing manner.
> For example, in Table 1, below the row "Our Results", there are:
> - VQ-VAE
> - VQ-VAE with EM
> - VQ-VAE + distillation
> - VQ-VAE with EM + distillation
>
> The "VQ-VAE" is not the proposed model, correct?
> My understanding is that the proposal is a VQ-VAE solved via soft EM, which corresponds to "VQ-VAE with EM".
> <<<
>
> Yes VQ-VAE is not the proposed model, although we report it in "Our Results" because the implementation is different from Kaiser et al in two crucial aspects 1) No attention to source sequences for the discrete latents 2) Product Quantization (PQ) which the authors of Kaiser et al call DVQ is not being used. Hence we also report it in "Our Results".
>
> >>>
> Third, a paragraph "Robustness of EM to Hyperparameters" is mis-leading.
> The figure 3 does not show the robustness against a hyperparameter.
> It shows the BLEU against the number of "samples" (in fact, there is no explanation about what the "samples" means).
> I think hyperparameters are model constants such as the learning rate of the SGD, alpha-beta params for Adam, dimension of hidden units, number of layers, etc. The number of samples are not considered as a model hyperparameter; it's a dataset property.
> >>>
>
> The number of samples used for EM training of VQ-VAE is a hyperparameter, how is it a property of the dataset? You are free to choose any number of samples regardless of the dataset.
>
> >>>
> The figure 5 shows the reconstructed images of the original VQ-VAE and the proposed VQ-VAE with EM.
> However, there is no explanation which hyperparameter is tested to assess "the robustness to hyperparameters".
> <<<
>
> Our apologies, this should be robustness to initialization of the codebook. VQ-VAE/K-means is much more sensitive to a good initialization as compared to EM.
>
> >>>
> Fourth, there is no experimental report on the image reconstructions (with CIFAR and SVHN) in the main manuscript.
> In fact, there is a short paragraph that mentions about the SVHN results,
> but it only refers to the appendix.
> I think appendix is basically used for additional results or proofs, that are not essential for the main message of the paper.
>
> However, performance in the image reconstruction is one of the main claims written in the abstract, the intro, etc.
> So, the authors should include the image reconstruction results in the main body of the paper.
> Otherwise, claims about the image reconstructions should be removed from the abstract, etc.
> >>>
>
> We have removed all image references from the main section and now only report it in the Appendix. We hope this helps improving the quality and clarity of the main paper.

---

> > ### Author Response · Authors · 2018-12-03
> > **Updated paper**
> >
> > R3, we believe have addressed your concerns and clarified some of your points. Do you have an updated impression of our paper? Thanks for your consideration.

---

> > > ### Comment · AnonReviewer3 · 2018-12-06
> > > **Thank you for your detailed feedback.**
> > >
> > > I'm happy to see the revised manuscript tightening its focus on the NMT. I think this makes it easier for readers to catch the core message of the paper.
> > > I also observed some of confusing remarks in the experiments are improved, this is another plus.
> > >
> > > Yet I still feel some difficulties in experimental section readings.
> > > In P.7,
> > >
> > > """
> > > The results are summarized in Table 1. Our implementation of VQ-VAE achieves a significantly
> > > better BLEU score and faster decoding speed compared to Kaiser et al. (2018).
> > > """
> > >
> > > The proposed model ("VQ-VAE" in table 1, I suppose) is compared with the results of (Kaiser+, 2018), but no numbers of (Kaiser+, 2018) are reported in the manuscript body, if I correctly read the paper. I propose to provide these numbers then the readers can verify these claims instantly.
> > >
> > > I also think it is better to emphasize the difference of your "VQ-VAE" and the implementation of the (Kaiser+, 2018) in more easier-to-understand-visually manner. for example, using a table?
> > > I'm afraid that it is still difficult for readers to understand why "VQ-VAE" is included in the "our results", without your detailed feedback comments.

---

> > > > ### Author Response · Authors · 2018-12-06
> > > > **Thanks for your attention**
> > > >
> > > > We thank the reviewer for carefully reading our updated manuscript and for raising their score. We acknowledge that it may be difficult for the reader to grasp the difference in implementation of VQ-VAE from Kaiser et al. We have now separated out the two VQ-VAE results in Table 1 in the Experiments section. It will be reflected in the final draft if the paper is accepted.

---

### Official Review · AnonReviewer1 · 2018-10-30
**A soft-EM training algorithm for vector-quantized autoencoders**

**Rating:** 3
**Confidence:** 4

**Review:**

Summary:

This paper presents a new training algorithm for vector-quantized autoencoders (VQVAE), a discrete latent variable model akin to continuous variational autoencoders.
The authors propose a soft-EM training algorithm for this model, that replaces hard assignment of latent codes to datapoints with a weighted soft-assignment.

Overall the technical writing in the paper is sloppy, and the presentation of the generative model takes the form of an algorithmic description of the training algorithm, rather than being a clear definition of the generative model itself.

The technical presentation of the work by the authors starts only at page 5 (taking less than a full page), after several pages of imprecise presentation of previous and related work. The paper could be significantly improved by making this preceding material more concise and rigorous.

Quantitative experimental evaluation is limited to a machine translation task, which is rather uncommon in the literature on generative latent variable models. I would expect evaluation in terms of held-out data log-likelihood (ie bits-per-dimension) used in probabilistic generative models, and possibly also using measures from the GAN literature such as inception scores. Datasets that are common include CIFAR-10 and resized variants of the imagenet dataset.


Specific comments:

- Please adhere to the ICLR template bibliography style, which is far more readable than the style that you used.

- Figure 1 does not seem to be referenced in the text.

- The last paragraph of section 2.1 is unclear. It mentions a sampling a sequence of latent codes. The notion of sequentiality has not been mentioned before, and it is not clear what it refers to in the context of the model defined so far up to that point.

- The technical notation is very sloppy.
* In numerous places the paper refers to the joint distribution P(x1,…,x_n, z1, …, zn) without defining that the distribution factorizes across the samples (xi,zi), and without specifying the forms of p(zi) and p(xi|zi).
* This makes that claims such as “computing the expectation in the M step (Equation 11) is computationally infeasible” are not verifiable.

- Please be clear about how much is gained by replacing the exact M-step with a the one based on the samples from the posterior computed in the E-step.

- What is the reason to decode the weighted average of the embedding vectors, rather than decoding all of them, and updating the decoder in a weighted manner?

- reference 14 for Variational autoencoders is incorrect, please use the following citation instead:
@InProceedings{kingma14iclr,
  Title                    = {Auto-Encoding Variational {B}ayes},
  Author                   = {D. Kingma and M. Welling},
  Booktitle                = {{ICLR}},
  Year                     = {2014}
}

- The related work section (4) provides a rather limited overview of relevant related work.
Half of it is dedicated to recent advances in machine translation, which does not bear a direct connection to the technical material presented in section 3.

- There is no justification of using *causal* self-attention on the source embedding, is this a typo?

- As for the experimental evaluation results: it seems that distillation is a much more critical factor to achieve good performance than the proposed EM training of the VQ-VAE model. Unfortunately, this fact goes unmentioned when discussing the experimental results.

- What is the significance of the observed differences in BLEU scores? Please report average performance and standard deviations over several runs with randomized parameter initialization and batch scheduling.

- It seems that the tuning of the number of discrete latent codes (table 2 in appendix) and other hyper-parameters (table 3 in appendix) was done on the test set, which is also used to compare to related work. A separate validation set should be used for hyper parameter tuning in machine learning experiments.

- It seems that all curves in figure 3 collapse from about 45 BLEU to values around 17 BLEU, why is this? The figure is hard to read since poor quality, and curves that are superposed.

---

> ### Author Response · Authors · 2018-11-09
> **Reply to Reviewer 1**
>
> We thank the reviewer for taking the time to read our paper. Below we address the specific points raised by the reviewer:
>
> >>>
> Overall the technical writing in the paper is sloppy....
> <<<
>
> In this work, we improve upon VQ-VAE to learn shorter latent representations of a target sentence in order to speed up MT, rather than to train a generative model. We achieve considerable speedup in decoding state of the art NMT models without much loss in BLEU (a universally accepted metric for translation quality), which has powerful implications for real world, production level MT systems. While evaluating the improvements of our training for generative modeling is interesting, our focus is on using VQ-VAE for a practical task.
>
> Moreover, we have now added a paragraph on the generative process (Page 3). We hope that this will clarify some of the content. We welcome the reviewer to share what they think is "sloppy" and "imprecise", and what would help us further improve the content of the paper.
>
> >>>
> The technical presentation of the work by the authors starts only at page 5...
> <<<
>
> Our goal is to use the autoencoder from VQ-VAE as a tool to compress the target sentence for fast decoding. We therefore chose to focus on the part of the algorithm, describing it's connection to hard-EM and our improvements on it using EM. We would appreciate concrete suggestions to improve the content.
>
> >>>
> Quantitative experimental evaluation is limited to a machine translation task...
> <<<
>
> The main focus of our work is to design a better non-autoregressive machine translation model and which is an area of active research (see for e.g., [1, 2, 3, 4]). None of those works evaluate their proposed method on datasets other than machine translation because the goal of their work is non-autoregressive MT. We do not care about generative modeling of images with VQ-VAE because plenty of other models do it much better (for e.g., a GAN/VAE/PixelCNN++).
>
> The keywords of our paper states: "machine translation, vector quantized autoencoders, non-autoregressive, NMT", while the TL;DR of our submission is "Understand the VQ-VAE discrete autoencoder systematically using EM and use it to design non-autogressive translation model matching a strong autoregressive baseline."
>
> >>>
> - The related work section (4) provides a rather limited overview of relevant related work...
> <<<
>
> Again, the main aim of our work is to speed up the decoding for real world Neural Machine Translation (NMT) systems, which is an active area of research (see e.g., [1, 2, 3, 4]). We have focussed on generative models that are practically relevant to non-autoregressive NMT and because of page limitations we have not been able to include every paper on generative modeling. If we have missed relevant references we would appreciate if the reviewer would let us know what they are.
>
> [1] https://openreview.net/forum?id=B1l8BtlCb
> [2] http://proceedings.mlr.press/v80/kaiser18a/kaiser18a.pdf
> [3] https://openreview.net/forum?id=r1gGpjActQ
> [4] https://arxiv.org/abs/1802.06901

---

> > ### Author Response · Authors · 2018-11-09
> > **Reply to Reviewer 1 continued**
> >
> > Continued from above:
> >
> > >>>
> > - There is no justification of using *causal* self-attention...
> > <<<
> >
> > Attention to the source embeddings is a natural and justified way to inform the discrete latents (see e.g., [2]). Also, the attention to source sequences for generating the discrete latents from the targets is not causal. The only causal attention layers are for encoding the inputs and in the autoregressive decoder from the latents.
> >
> > >>>
> > - As for the experimental evaluation results: it seems that distillation...
> > <<<
> >
> > In page 7 of the current draft (and page 6 of the original submission), we say "Additionally, we see a large improvement in the performance of the model by using sequence-level distillation (12), as has been observed previously in non-autoregressive models (6; 16)." We have also added a sentence to this effect in the conclusion in the updated draft.
> >
> > >>>
> > - What is the significance of the observed differences in BLEU scores? ...
> > <<<
> >
> > We point the reviewer to [1, 2, 3, 4] which are the current state-of-the-art literature on non-autoregressive machine translation. None of these works report average or std devs on several runs, instead they select the best hyperparameter from a validation set and report the result of this model on a held out test set (which is a perfectly valid thing to do).
> >
> > >>>
> > - It seems that the tuning of the number of discrete latent codes...
> > <<<
> >
> > The optimal hyperparameters are selected on the validation set (WMT'13) while the reported results are on the held out WMT'14 test set. This is standard practice in the NMT literature. We have made this more explicit in the latest draft.
> >
> > >>>
> > - It seems that all curves in figure 3 collapse from about 45 BLEU...
> > <<<
> >
> > We have made this figure larger so that it is easier to read. The figure is intended to show the robustness of the EM runs vs the VQ-VAE runs: the collapsed curve is  a VQ-VAE run with bad initialization, while the other superimposed curves are different EM runs of the same configuration with various values of the number of samples.
> >
> > [1] https://openreview.net/forum?id=B1l8BtlCb
> > [2] http://proceedings.mlr.press/v80/kaiser18a/kaiser18a.pdf
> > [3] https://openreview.net/forum?id=r1gGpjActQ
> > [4] https://arxiv.org/abs/1802.06901

---

> > > ### Author Response · Authors · 2018-12-03
> > > **Updated paper**
> > >
> > > R1, we believe have addressed your concerns and clarified some of your points. Do you have an updated impression of our paper? Thanks for your consideration.

---

### Official Review · AnonReviewer2 · 2018-11-01
**Training procedure for VQ-VAE is equivalent to the EM algorithm**

**Rating:** 7
**Confidence:** 4

**Review:**

General:
The paper presents an alternative view on the training procedure for the VQ-VAE. The authors have noticed that there is a close connection between the original training algorithm and the well-known EM algorithm. Then, they proposed to use the soft EM algorithm. In the experiments the authors showed that the soft EM allows to obtain significantly better results than the standard learning procedure on both image and text datasets.

In general, the paper shows a neat link between the well-known EM algorithm and the learning method for the VQ-VAE. I like the manner the idea is presented. Additionally, the results are convincing. I believe that the paper will be interesting for the ICLR audience.

Pros:
+ The connection between the EM algorithms and the training procedure for the VQ-VAE is neat.
+ The paper is very well written, all concepts are clear and properly outlined.
+ The experiments are properly performed and all results are convincing.

Cons:
- The paper is rather incremental, however, still interesting.
- The quality of Figure 1, 2 and 3 (especially Figure 3) is unacceptable.
- There is a typo in Table 6 (row 5: V-VAE → VQ-VAE).
- I miss two references in the related work on training with discrete variables: REBAR (Tucker et al., 2017) and RELAX (Grathwohl et al., 2018).
- The paper style is not compliant with the ICLR style.

--REVISION--
I would like to thank authors for their effort to improve quality of images. In my opinion the paper is nice and I sustain my initial score.

---

> ### Author Response · Authors · 2018-11-08
> **Reply to reviewer 2**
>
> We thank the reviewer for taking the time to read our paper and for the useful comments to help improve our presentation! We have increased the resolution of the images by moving some of them to the appendix, and hope that fixes the visibility issue for the figures. We have also fixed the typo - thanks for pointing it out! We have added the two references pointed out and have also fixed the bibliography style to be the ICLR style. Please let us know if we can improve anything else.

---

### Official Review · AnonReviewer4 · 2018-11-15
**interesting parts, but needs more rigour**

**Rating:** 5
**Confidence:** 4

**Review:**

This paper discusses VQ-VAE for learning discrete latent variables, and its application to NMT with a non-autoregressive decoder to reduce latency (obtained by producing a number of latent variables that is much smaller than the number of target words, and then producing all target words in parallel conditioned on the latent variables and the source text). The authors show the connection between the existing EMA technique for learning the discrete latent states and hard EM, and introduce a Monte-Carlo EM algorithm as a new learning technique. They show strong empirical results on EN-DE NMT with a latent Transformer (Kaiser et al. (2018)).

The paper is clearly written (excepting the overloaded appendix), and the individual parts of the paper are interesting, including the link between VQ-VAE training and hard EM, the Monte-Carlo EM, and strong empirical results. I'm less convinced that the paper as a whole delivers on what it promises/claims.

The first contribution of the paper is that it shows a simple VQ-VAE to work well on the EN-DE NMT task, in contrast to the results by Kaiser et al. (2018). The paper attributes this to tuning of the code-book, but the results (table 3) seem to contradict this, with a code-book size of 2^16 even slightly better than the 2^12 that is used subsequently. The reason for the performance difference to Kaiser et al. (2018) remains opaque. While interesting, the empirical effectiveness of Monte-Carlo EM is a bit disappointing, achieving +0.3 BLEU over the best configuration for EN-DE (after extensive hyperparameter tuning, seen in table 4), and -0.1 BLEU on EN-FR. Monte-Carlo EM also seems very sensitive to hyperparameters, namely the sample size (tables 4,5), contradicting the later claim that EM is robust to hyperparameters. The last claimed contribution (using denoising techniques) is hidden in the appendix, an application of an existing technique, and not compared to knowledge distillation (another existing technique).

I'd like to see some of the results in the paper published eventually. However, the claims need to better match the empirical evidence, and for a paper that has "better understanding" in the title, I'd like to gain a better understanding of the differences to Kaiser et al. (2018) that make VQ-VAE fail for them, but not in the present case.

+ clearly written paper
+ interesting, novel EM algorithm for VQ-VAE
+ strong empirical results on non-autoregressive NMT

- the strong performance of the VQ-VAE baseline remains unexplained, and the claimed explanation contradicts empirical results.
- the new EM algorithm gives relatively small improvements, with hyperparameters that were likely selected based on test set scores .
- most of the empirical gain is attributable to knowledge distillation, which is not a novel contribution

---

> ### Comment · AnonReviewer4 · 2018-11-15
> **reading other reviews/comments**
>
> This was an extra review requested after the end of the official review period; now looking at the other reviews and replies, I can see that the question as to whether hyperparameters were optimized on the test set was already addressed. I stand by the comment that obtaining this small improvement required extensive hyperparameter tuning, which devalues it slightly.

---

> > ### Author Response · Authors · 2018-11-15
> > **Reply to Reviewer 4**
> >
> > We thank the reviewer for a careful reading of our paper and for their thoughtful review. Below we address the specific points raised by the reviewer:
> >
> > >>>
> > The first contribution of the paper is that it shows a simple VQ-VAE to work well on the EN-DE NMT task, in contrast to the results by Kaiser et al. (2018)...
> > <<<
> >
> > The main difference between the setup of Kaiser et al (2018) and the current work is the point "Attention to Source Sentence Encoder" in the Analysis section. The discrete latents in Kaiser et al (2018) are a function ae(x, y) where x is the input sequence and y is the target sequence. The dependence on x is in the form of attention layers. This makes it a much more complex function to learn and the authors of that work report that VQ indeed did not work for them, and so they had to resort to Product Quantization (referred to as DVQ in their work) with multiple codebooks to get a good result. We found that the attention to source sequence x to be an unnecessary complication, and so our latents are just a function ae(y), where y is the target sequence.
> >
> > We do not attribute this to tuning the code-book size, we apologize for the misunderstanding. The robustness of EM is in the case when the latents ae(x, y) are also a function of x, see Figure 5 in the Appendix (" Comparison of VQ-VAE (green curve) vs EM with different number of samples (yellow and blue curves) on the WMT’14 English-German translation dataset with a codebook size of 2 14, with the encoder of the discrete autoencoder attending to the output of the encoder of the source sentence as in Kaiser et al. (2018).") The optimization problem is much harder in this case and we see that the VQ runs collapse while various versions of EM (with different number of samples) still give a good result.  The EM version does depend on the number of samples, but is much more stable compared to VQ even when the latents are a function of x.
> >
> > We apologize if the "Attention to source sentence encoder" was not adequately clear: we had a statement to the effect of "Also, removing this attention step results in more stable training particularly for large code-book sizes, see e.g., Figure 3.", but it unfortunately seems to have got lost in a revision..
> >
> > >>>
> > The last claimed contribution (using denoising techniques) is hidden in the appendix...
> > <<<
> >
> > Denoising autoencoders as used by Lample et al., were used in the context of learning better initial representations for unsupervised MT. We found that applying it to the context of discrete autoencoders like VQ-VAE can give some improvements
> >  in larger datasets like En-Fr. For En-De denoising VQ-VAE did not give us any improvement over VQ-VAE. We do not claim we invented denoising autoencoders, we write:
> >
> > "On the larger English-French dataset, we show that denoising discrete autoencoders gives us a significant improvement (1.0 BLEU) on top of our non-autoregressive baseline (see Section D)"
> >
> > >>>
> > I'd like to see some of the results in the paper published eventually...
> > <<<
> >
> > We hope our first paragraph addresses the question of why VQ-VAE did not work in Kaiser et al. without product quantization, but worked in our case. We have made this more explicit in the latest version. Also note that all our VQ-VAE runs for MT do not have the attention to source sequence x, except Figure 5 where we explicitly mention this.
> >
> > >>>
> > - the strong performance of the VQ-VAE baseline remains unexplained, and the claimed explanation contradicts empirical results.
> > <<<
> >
> > We hope that the previous paragraphs and the new draft addresses this concern.
> >
> > >>>
> > - the new EM algorithm gives relatively small improvements, with hyperparameters that were likely selected based on test set scores .
> > <<<
> >
> > The hyperparameters were selected on WMT'13 and the results are reported on WMT'14. EM gives small improvements with knowledge distillation, because the optimization problem is much easier in this case. When the optimization problem is harder we see more gains from EM:
> >
> > 1) In the setting when the latents are informed by the source sequence x, EM is much more stable than VQ-VAE (Figure 5)
> > 2) In the case when knowledge distillation is not used it gives a gain of +1.0 BLEU
> > 3) When the hidden dimension is smaller (256 or 384) instead of 512, we see gains of +1.3 BLEU and +0.6 BLEU respectively.
> >
> > >>>
> > - most of the empirical gain is attributable to knowledge distillation, which is not a novel contribution
> > <<<
> >
> > That is a valid point, and we did indeed find knowledge distillation to be very important for good performance for NMT in addition to removing the attention to source sequence x.

---

> > > ### Author Response · Authors · 2018-12-03
> > > **Updated paper**
> > >
> > > R4, we believe have addressed your concerns and clarified some of your points. Do you have an updated impression of our paper? Thanks for your consideration.

---

> > > > ### Comment · AnonReviewer4 · 2018-12-05
> > > > **minimal update**
> > > >
> > > > thank you for drawing attention to your updated draft. While your comment (and the added sentence in 5.1.1) clarifies the difference between Kaiser et al. (2018) and this work, the presentation in the paper is still misleading (the introduction does not refer to this architecture difference, and still attributes performance differences to "tuning for the code-book size".
> > > >
> > > > I don't believe the new draft made any other changes that would update my impression.

---

> > > > > ### Author Response · Authors · 2018-12-06
> > > > > **Thanks for your attention**
> > > > >
> > > > > We thank the reviewer for reading our updated manuscript and for their feedback. We acknowledge that we missed this sentence in the introduction, since we focused on updating the experimental section and the writing therein as per the comments of R3. We will definitely go over the manuscript carefully and make the presentation more clear. Do you have any more specific aspects of the presentation that you would like to be improved ?

---

### Meta-Review · Area_Chair1 · 2018-12-15
**a reasonable method but empirical evidence is questionable**

**Confidence:** 4
**Recommendation:** Reject

**Metareview:**

Strengths:

- well-written
- strong results for non-autoregressive NMT
- a novel soft EM version of VQ-VAE

Weaknesses:

-  as pointed out by reviewers, the improvements are mostly not due to the VQ-VAE modification rather due to orthogonal (and not interesting) changes e.g., knowledge distillation. If there is a genuine contribution of VQ-VAE, it is small and required extensive parameter selection

-  the explanations provided in the paper do not match the empirical results

Two reviewers criticize the experiments / experimental section: rigour / their discussion.  Overall, there is nothing wrong with the method but the experiments are not showing that the modification is particularly beneficial.  Given these results and also given that the method is not particularly novel (switching from EM to Soft EM in VQ-VAE), it is hard for me to argue for accepting the paper.